# Bistatic Forward ISAR with DVB-T Transmitter of Opportunity

**DOI:** 10.3390/s21196662

**Published:** 2021-10-07

**Authors:** Andon Dimitrov Lazarov, Todor Pavlov Kostadinov

**Affiliations:** 1Nikola Vaptsarov Naval Academy, 9002 Varna, Bulgaria; 2Aerospace Engineering Faculty, K.N. Toosi University of Technology, Tehran 15119-43943, Iran; 3Organic Department, Bourgas Prof. Assen Zlatarov University, 8010 Bourgas, Bulgaria; kostadinov_todor@btu.bg

**Keywords:** bistatic forward ISAR, DVB-T waveform, BFISAR signal model, BFISAR image reconstruction

## Abstract

The radar geometry defined by a spatially separated transmitter and receiver with a moving object crossing the baseline is considered as a Bistatic Forward Inverse Synthetic Aperture Radar (BFISAR). As a transmitter of opportunity, a Digital Video Broadcast-Terrestrial (DVB-T) television station emitting DVB-T waveforms was used. A system of vector equations describing the kinematics of the object was derived. A mathematical model of a BFISAR signal received after the reflection of DVB-T waveforms from the moving object was described. An algorithm for extraction of the object’s image including phase correction and two Fourier transformations applied over the received BFISAR signal—in the range and azimuth directions—was created. To prove the correctness of mathematical models of the object geometry, waveforms and signals, and the image extraction procedure, graphical results of simulation numerical experiments were provided.

## 1. Introduction

Synthetic Aperture Radar (SAR) and Inverse Synthetic Aperture Radar (ISAR) are microwave-imaging systems with military and civil applications functioning under all meteorological conditions, day and night [1,2]. To realize a cross-range or azimuth resolution, SAR uses the SAR carrier’s displacement in respect of the observed object, whereas ISAR uses the object’s movement in respect of a stationary system of observation. To achieve a range resolution, both systems utilize high-information wideband waveforms. Bistatic Synthetic Aperture Radar and Multi-static Synthetic Aperture Radar (BSAR/MSAR) with spatially separated transmitters, receivers, and objects are a subclass of SAR/ISAR systems. Nowadays, BSAR/MSAR technologies and corresponding signal processing methods and algorithms are experiencing a renaissance in their development [2,3,4,5,6] due to their exceptional properties of stealth target detection, covert surveillance, detection, imaging, recognition, and tracking, as well as using radio-emitters of opportunity such as satellite communication transmitters, GPS and SAR transmitters, other space systems and bodies such as the sun, pulsars, etc.

The Bistatic SAR and Multistatic SAR (BSAR/MSAR) systems have stimulated development of new signal and image processing techniques. Original theoretical conceptions and their experimental verifications were presented in [3,4]. System model’s constructions, imaging algorithms, mission designs, as well as interferometric SAR considered as a bistatic system, are the focus. The spatial separation of the transmitter and receiver in the BSAR topology makes it a promising and useful supplement to classical monostatic SAR/ISAR systems. An integrated time and phase synchronization strategy for a multichannel spaceborne-stationary BSAR system was proposed in [5].

The problem of migration through resolution cells induced by the rotational motion of a large space target, and sparse high-resolution imaging based on compressed sensing in high-resolution BSAR systems were discussed in [6].

Signal processing algorithms as remote sensing instruments in passive bistatic SAR with navigation satellites (e.g., GPS, GLONASS, or Galileo) as transmitters of opportunity were discussed in [7]. Signal synchronization and image formation algorithms were described. Image products such as two-color multi-view, image quality, image characteristics, and coherent change detection formed from BSAR signals during the validation stage were presented in [8]. A ground moving target detection (GMTD) method based on joint clutter cancellation in the echo-image domain for forward-looking BSAR to achieve effective GMTD in heterogeneous clutter was suggested in [9].

For human positioning and activity recognition, millimeter-wave radar, as an emerging technique, is exceptionally appropriate. In contrast to traditional sensors and radars, millimeter-wave radars give detailed information on objects from the range domain to the Doppler domain. The short wavelength allows millimeter-wave radars to achieve a high resolution and a small antenna size, but also makes them prone to noise. In [10], a system framework for human detection and tracking using millimeter-wave radars was designed.

A k-nearest neighbors approach to the design of radar detectors was considered in [11]. The idea is to start with either raw data or well-known radar statistics as a feature vector used in the definition of the decision rule. Simulation experimental results obtained using real clutter recordings were provided to illustrate the behavior of detectors derived using the proposed approach.

Constant false alarm rate (CFAR) detection algorithms have been the focus of the BSAR/MSAR systems’ implementation. A generalized likelihood ratio test-based adaptive detection algorithm defined in the CFAR feature space, where observed data are mapped to clusters, was analytically described in [12]. Linear and nonlinear detection algorithms with robust selective properties were proposed.

An along-track multistatic SAR system with a high azimuth resolution maintaining a large swath width was discussed in [13]. A SIMO (Single-Input Multiple-Output) construction with one transmitting sensor and multiple receiving sensors providing low azimuth ambiguity and a recombination gain close to the theoretical one was analyzed.

Automotive radar with its multiple sensors is a multistatic radar with specific requirements such as high resolution, low hardware cost, and small size satisfied by a multiple-input, multiple-output (MIMO) radar configuration. It provides a high angular resolution with relatively small numbers of antennas. This technology is embedded in the current-generation automotive radar for advanced driver-assistance systems, as well as in next-generation high-resolution imaging radar for autonomous driving. A review of MIMO radar basics that highlights the features of the automotive radar and provides important theoretical results for increasing the angular resolution was suggested in [14].

Hardware architecture, video signal processing, and field experimental results of a multirotor Video SAR imaging system with ultrahigh resolution were provided in [15]. A Field-Programmable Gate Array (FPGA)-based unified signal processing architecture was used to accelerate the generation of massive Video SAR sequences in terms of both circular spotlight and strip map modes.

An analytical model based on the Kirchhoff and geometrical optics approximation for evaluation of the electromagnetic field, single- and multiple-bounce scattering from a composite target in a generic bistatic radar configuration, was derived in [16]. Machine learning and deep learning with sensors data fusion are widely used in computer vision, natural language understanding, and data analytics. Hence, machine-learning algorithms are appropriate to solve identification problems that inherently include multi-modal data [17].

Modern radar systems, especially SAR systems, pose high requirements in respect of accuracy, robustness, and real-time operability in increasingly complex electromagnetic environments. Classical radar signal and image processing methods experience limitations when meeting such requirements, particularly in the area of target classification. Machine-learning, especially deep learning, can be applied to solve problems such as signal and image processing. A comprehensive, structured, and reasoned literature overview of machine learning, signal, and image processing techniques was provided in [18]. The survey disclosed the essence and application of machine learning techniques in SAR signal and image processing.

Contemporary radar surveillance systems consist of highly complex tracking, sensor data fusion, and identification algorithms tracking the trajectories of moving objects. These algorithms are embedded in a real-time middleware with a straightforward processing chain according to the fusion model. New technologies such as distributed data processing and machine learning disclose new possibilities for surveillance systems. An overview of how these technologies, in combination with the big data of trajectories, can be integrated into existing surveillance systems and how machine learning can help to improve situational awareness was suggested in [19].

Artificial intelligence due to recent breakthroughs in deep learning changes applications and services in all scientific domains, including radar technology. Deep learning relies on large amounts of training data that are mostly generated at the network edge by Internet of Things devices and sensors. Bringing the sensed data from the edge of a distributed network to a centralized cloud is often infeasible because of the massive data volume, limited network bandwidth, and real-time application constraints. A framework for data fusion and artificial intelligence processing at the edge was proposed in [20]. A comparative discussion of different data fusion and artificial intelligence models and architectures was provided. Multiple levels of fusion and different types of artificial intelligence were also discussed.

The application of artificial intelligence in target surveillance based on radar sensors is enabled through today’s computational capacities. A survey of past approaches and recent hot topics in the area of artificial intelligence applicable for target surveillance with radar sensors that reveals new potential for the development of novel approaches in radar research and practice was presented in [21]. The focus is on clutter identification, target classification, and target tracking, which are not only of great importance for an adequate operation of radar applications, but are also well suited for the use of artificial intelligence.

Analysis of the signal classification performances of various classifiers for deterministic signals under the additive white Gaussian noise in a wide range of signal-to-noise ratio (SNR) levels was performed in [22]. The matched filter bank classifier, convolutional neural networks, and the minimum distance classifier using spectral-domain features for the radar signal classification were derived. An unsupervised generative convolutional neural network approach for interferometric phase filtering and coherence estimation was suggested in [23].

An original approach in SAR/BSAR signal and image processing known as Compressed Sensing (CS) based on sparse decomposition was applied for improving target detection, parameter estimation, and identification [24,25]. A simulation of inverse synthetic aperture radar (ISAR) imaging of a space target and reconstruction under sparse sampling via CS was developed in [26]. Based on the linear dependence between the LFM rate of an ISAR signal in cross-range and the slant range, a parametric sparse representation method for ISAR imaging of rotating targets was proposed in [27]. CS-based models for inverse synthetic aperture radar (ISAR) imaging, which can sustain strong clutter noise and provide high-quality images with extremely limited measurements, were proposed in [28,29].

Based on sparsity in the ISAR azimuth frequency domain on the cross-range direction, a Fourier basis was applied as a sparse basis to achieve high cross-range resolution imaging by using a compressed sensing method. An improved Fourier basis for ISAR signal’s sparse representation to achieve robust recovery performance via CS was presented in [30]. Bistatic forward inverse synthetic aperture radar systems are a subclass of bistatic inverse synthetic aperture radars and characterized by a bistatic angle of the transmitter-object-receiver close or equal to 180° [31,32]. BFISAR geometry and kinematics can be considered as scenarios with a spatially separated stationary transmitter, stationary receiver, and an object crossing the baseline transmitter-receiver. The electromagnetic field emitted by the transmitter antenna is reemitted by the object and scatters forward to the receiver. Received signals are registered in two orthogonal coordinates, with the range measured along the line of sight and the cross-range or azimuth measured along the length of the inverse synthetic aperture [31,32,33].

BFISAR models, imaging methods, and algorithms have attracted considerable attention in the field of radar research in the last twenty years. A time–frequency approach for target imaging applied in forward scattering radar (FSR) using DVB-T transmitters of opportunity operating in a Very High Frequency (VHF) band was presented in [34].

A compressive sensing algorithm for BISAR imaging with DVB-T waveforms was discussed in [35]. BFISARs are exploited for air traffic monitoring and airport surveillance. As an example, a passive radar system for airport surveillance using a signal of opportunity from DVB-S (Digital Video Broadcasting—Satellite) was presented in [36]. Bistatic forward scattering radar detection and imaging was analyzed in [37]. An electromagnetic model of scattering response measured by FSR in far and near fields for surveillance air-traffics was suggested in [38]. Moving target shadows using passive forward scatter radar systems were studied in [39,40,41].

From the authors’ point of view, the available literature lacks a detailed description of the 3-D geometry and kinematics of the BFISAR scenario, the signal model based on the DVB-T waveform, and the 3-D geometry of the observed object. There is no analytical structure of the object’s image extraction process or its interpretation, based on which the algorithm for signal processing and image reconstruction is built.

The aim of the present work was to define the geometrical and kinematical features of the BFISAR system with a transmitter of opportunity, emitting BVB-T waveforms, to synthesize the BFISAR signal model based on the DVB-T waveform, and based on the vector description of the scenario and signal model, to build an algorithm for extraction of the BFISAR image.

The paper is structured in the following manner. Section 2 presents BFISAR geometric and kinematic vector equations describing the positions of the transmitter, receiver, and object. Section 3 describes stages of the algorithm for extraction of the object image. Section 4 presents the source data and graphical results of simulation experiments. In Section 5, concluding remarks are drawn.

## 2. BFISAR Geometrical and Kinematical Equations

Consider a BFISAR scenario with a DVB-T stationary transmitter ***T***, receiver ***R***, and moving object presented in a Cartesian coordinate system Oxyz, as shown in Figure 1. The object’s geometry depicted in its own coordinate system O’XYZ is presented as an assembly of reference points placed in nodes of a regular grid. The reference points, whose intensities describe the object’s geometry, are scattering points. Intensities of the remaining reference points are zero or noise-defined. The object is moving near the base line, while the transmitter and receiver can be placed on hills, mountain heights, or roofs of high buildings near the area of surveillance.

Vectors $R^{s}$ and $R^{r}$ define the positions of the transmitter and receiver, respectively, in a coordinate system Oxyz. The line ***TR*** is the baseline. The distance vector $R_{ijk}$ defines the position of the ijk-th scattering point in the coordinate system O’XYZ. The vector $R_{00’}(p)$ defines the position of the object’s mass-center at the p-th-emitted DVB-T segment. The geometrical and kinematical ratios in Figure 1 can be described by the vector equations as follows.

The vector distance of the transmitter–object’s mass center:(1)$$R^{s}(p)=R^{s}-R_{00’}(p)=R^{s}-R_{00’}(0)-V.p.T_{p}$$
where V is the object’s vector velocity, $p=\bar{0,\;N-1}$ is the index of the emitted segment, N is the total number of DVB-T segments emitted during bistatic inverse aperture synthesis, and $T_{p}$ is the segments’ time repetition period for aperture synthesis.

The vector distance of an object’s mass center–DVB-T receiver:(2)$$R^{r}(p)=R^{r}-R_{00’}(p)=R^{r}-R_{00’}(0)-V.p.T_{p}$$

The vector distance of a DVB-T transmitter–ijk-th object’s scattering point:(3)$$R^{s}{}_{ijk}(p)=R^{s}(p)+AR_{ijk}=R^{s}-R_{00’}(0)-V.p.T_{p}+AR_{ijk},$$
where **A** is the Euler transformation matrix.

The vector distance of an ijk-th object’s scattering point–DVB-T receiver:(4)$$R^{r}{}_{ijk}(p)=R^{r}(p)-AR_{ijk}=R^{r}-R_{00’}(0)-V.p.T_{p}-AR_{ijk}$$

The total distance of a DVB-T transmitter-ijk-th object’s scattering point–DVB-T receiver is defined by:(5)$$\left|R_{ijk}(p)\right|=\left|R^{s}{}_{ijk}(p)\right|+\left|R^{r}{}_{ijk}(p)\right|$$

Expression (5) is applied in the determination of the signal time delays from the object’s scattering points and synthesis of the BFISAR signal.

## 3. DVB-T Waveform and BFISAR Signal Synthesis

### 3.1. DVB-T Waveform Synthesis

DVB-T is a terrestrial-based digital video broadcasting system that transmits compressed digital audio, digital video, and other data in a Moving Picture Experts Group (MPEG) transport stream, using coded orthogonal frequency-division multiplexing (OFDM) modulation. The DVB-T television station emits waveforms, whose analytical description is based on the multiplication of the following complex exponential functions [42]:(6)$$s(t)=c_{m,l,k}.\psi_{m,l,k}(t).\exp(j2\pi f_{0}t),$$
where $f_{0}=c/\lambda_{0}$ is the central carrier frequency, $\lambda_{0}$ is the central wavelength, and *c* is the speed of light:

$m=\bar{0,\;\infty }$ stands for the DVB-T frame index;

$l=\bar{0,\;67}$ stands for the Orthogonal Frequency Division Multiplexing (OFDM) symbol index;

$k=\bar{-(K/2)+1,(K/2)}$ stands for the index of the carrier frequency, K=6817 denotes the total number of DVB-T carrier frequencies, and $c_{m,l,k}$ denotes a Quadrature Phase Shift Keying (QPSK) or Quadrature Amplitude Modulation (QAM) amplitude defined for the k-th carrier, l-th data symbol, and m-th frame.

The values of the exponential function $\psi_{m,l,k}(t)$ depend on the time intervals in which it is defined. If the current time t accepts values in the interval $t=\bar{\left(l+68.m\right).T_{S},\left(l+68.m+1\right).T_{S}}$, the exponential function $\psi_{m,l,k}(t)$ can be written as:(7)$$\psi_{m,l,k}(t)=\exp\left\{j2\pi \frac{k}{T_{U}}.\left[.t-\Delta -(l+68.m)T_{S}\right]\right\}$$
where $T_{S}=\Delta +T_{U}$ is the general symbol duration, $T_{U}=0.896$ m*s* is the symbol’s part duration, Δ=0.224 m*s* is the guard interval in the general symbol, and $T_{S}=1.12$ ms.

Otherwise, the exponential function $\psi_{m,l,k}(t)$ is equal to zero, i.e., $\psi_{m,l,k}(t)=0$.

A DVB-T waveform-synthesizing algorithm includes the following steps.
For m=0, l=0, $k=\bar{-(K/2)+1,(K/2)}$, compute $\psi_{m,l,k}(t)$. If k=K/2, then l=1.For m=0, l=1, $k=\bar{-(K/2)+1,(K/2)}$, compute $\psi_{m,l,k}(t)$, then l=l+1.When l=67, then m=m+1, go to step 2.

Denote $\tilde{t}=\Delta +(l+68.m)T_{S}$ as a slow time parameter measured on the azimuth direction. The complex DVB-T waveform can be written as
(8)$$s(t)=\exp\left\{j.2\pi .\left[\frac{k}{T_{U}}.\left(t-\tilde{t}\right)+f_{0}.t\right]\right\}$$
which can be rewritten as:(9)$$s(t)=\exp\left\{j.2\pi .\left[\left(f_{0}+k.\Delta f\right)t-k.\Delta f.\tilde{t}\right]\right\}$$
where $\Delta f={\left(T_{U}\right)}^{-1}$ is the carriers’ spacing.

### 3.2. BFISAR Signal Synthesis

Assume the object depicted in a 3-D regular grid with scattering points placed in its nodes is illuminated by DVB-T waveforms. Partial signals reflected from the target’s scattering points are superposed according to time delays defined by distances $R_{ijk}(p)$ and the light speed. The segments’ repetition period during the aperture synthesis can be defined by $T_{p}=\Delta .l.T_{S}$, where Δl is the interval between segments’ indices used. The total time of the aperture synthesis can be calculated by $\hat{T}=m\times \Delta .l.T_{S}$, where the parameters’ values, m and Δ.l, depend on the synthetic aperture length, object’s velocity, and realized azimuth resolution. For example, to realize an appropriate segments’ repetition period, symbols indexed by l=0, 5, 10,…, 65 for $m=\bar{0,\;9}$ frames can be registered and used for aperture synthesis.

The complex BFISAR signal reflected from the ijk-th scattering point can be expressed as:(10)$$S_{ijk}(k,p)=\mathrm{rect}\left(\frac{t-t_{ijk}}{T_{U}}\right)a_{ijk}.\exp\left\{j.2\pi .\left[\left(f_{0}+k.\Delta f.\right)\left(t-t_{ijk}\right)-k.\Delta f.\tilde{t}\right]\right\}$$
for $k=\bar{-(K/2)+1,(K/2)}$, and $p=\bar{0,\;N-1}$, where $a_{ijk}$ is the intensity of the ijk-th scattering point, and $t_{ijk}=t_{ijk}(p)=\frac{R_{ijk}(p)}{c}$ is the signal time delay from the ijk-th scattering point.

The rectangular function $\mathrm{rect}\frac{t-t_{ijk}}{T_{U}}$ determines the DVB-T symbol size in the synthesized BFISAR signal, and can be defined as:(11)$$\mathrm{rect}\frac{t-t_{ijk}}{T_{U}}=\left\{\begin{array}{l} 1\;\mathrm{if}\;-0.5\leq \frac{t-t_{ijk}}{T_{U}}<0.5\\ 0,\;\;\mathrm{otherwise} \end{array}\right.$$

The complex BFISAR signal reflected from the 3-D object can be written as:(12)$$S(k,p)=\sum_{i,j,k}\mathrm{rect}\left(\frac{t-t_{ijk}}{T_{U}}\right)a_{ijk}.\exp\left\{j2\pi .\left[\left(f_{0}+k.\Delta f.\right)\left(t-t_{ijk}\right)-k.\Delta f.\tilde{t}\right]\right\}.$$
for $k=\bar{-(K/2)+1,(K/2)}$ and $p=\bar{0,\;N-1}$. 

While BFISAR signal modeling, the current time *t* measured in the range direction is defined by the expression:(13)$$t=t_{ijk\min}(p)+\tau ,$$
where $t_{ijk\min}(p)=\frac{R_{ijk\min}(p)}{c}$ is the reference time defined by the signal time delay from the nearest scattering point, τ=k.ΔT is the fast time, $\Delta T=T_{U}.K^{-1}=0.1314$ μs is the time width of the DVB-T waveform’s sample, $\Delta f=T_{U}{}^{-1}=1.116$ kHz is the carrier spacing, and $\Delta F={\left(\Delta T\right)}^{-1}=7.6104$ MHz is the DVB-T sample’s frequency bandwidth.

Equation (12) describes the process of the BFISAR signal formation and can be interpreted as a spatial transformation of the 3-D image function $a_{ijk}$ into a 2-D BFISAR signal S(k,p), defined in discrete range *k* and cross-range *p* coordinates.

## 4. BFISAR Image Reconstruction Algorithm

The image extraction is the inverse operation to Equation (12). From the 2-D BFISAR signal S(k,p), a 2-D image can be extracted, i.e.,
(14)$$a(\hat{k},\hat{p})=\sum_{p=0,k=0}^{N-1,K-1}S(k,p)\exp\left\{-j2\pi .\left[\left(f_{0}+k.\Delta f.\right)\left(t-t_{ijk}\right)-k.\Delta f.\tilde{t}\right]\right\},$$
where $\hat{k}=\bar{0,\;K-1},\hat{p}=\bar{0,\;N-1}$ are the new unknown coordinate indices of the ijk-th scattering point.

To describe this process, the transformation of the 3-D image into a 2-D image, whose plane coincides with a signal plane, has to be considered. The 3-D coordinates of each scattering point are transformed into 2-D coordinates. The problem can be referred to a 3-D to 2-D projection, a design technique used to display the 3D object on the 2D surface. The exponential term in Equation (12) is a 3-D to 2-D projective operator. Consider the argument of the exponential term and denote as:(15)$$\hat{\Phi }(k,p)=2\pi \left\{.\left(f_{0}+k\Delta f\right)\left(t-\frac{R_{ijk}(p)}{c}\right)-k\Delta f.\tilde{t}\right\}.$$

Based on the 2-D Taylor expansion of $\hat{\Phi }(k,p)$ in the vicinity of the imaging point defined by signal coordinates k=0,p=0, the following expression can be written:(16)$$\hat{\Phi }(k,p)=\hat{\Phi }(0,0)+\frac{{\hat{\Phi }}^{\left(r\right)}(0,0)}{r!}{\left(p.T_{p}\right)}^{r}+\frac{{\hat{\Phi }}^{\left(q\right)}(0,0)}{q!}{\left(k.\Delta T\right)}^{q},$$
where $\hat{\Phi }(0,0)$ is the constant phase term that does not influence the imaging process; ${\hat{\Phi }}^{\left(r\right)}(0,0)$ is the r-th derivative of $\hat{\Phi }(k,p)$, where r=1,2,…, on the slow time $\tilde{t}$; ${\hat{\Phi }}^{\left(q\right)}(0,0)$ is the q-th derivative of $\hat{\Phi }(k,p)$, where q=1,2,…, on the fast time τ calculated for k=0,p=0.

In the case r=1, the first derivative of $\hat{\Phi }(k,p)$ for k=0,p=0 is equal to $2\pi .\frac{f_{0}}{c}.\frac{dR_{ijk}(0)}{d\tilde{t}}$, where $\frac{dR_{ijk}(0)}{d\tilde{t}}=v_{rijk}(0)$ defines the radial velocity of the scattering point. The linear term can be modified as follows.
(17)$${\hat{\Phi }}^{\left(1\right)}(0,0).(p.T_{p})=\frac{d\hat{\Phi }(00)}{d\tilde{t}}(p.T_{p})=-2\pi .\frac{f_{0}}{c}\frac{dR_{ijk}(0)}{d\tilde{t}}(p.T_{p})=-2\pi .\frac{c}{c.\lambda_{0}}\frac{dR_{ijk}(0)}{d\tilde{t}}(p.T_{p}).$$

Multiply and divide the expression with the period of the aperture synthesis and 2 (two), i.e.,
(18)$${\hat{\Phi }}^{\left(1\right)}(0,0).(p.T_{p})=-2\pi .\frac{2}{\lambda_{0}}v_{rijk}(0).\frac{\hat{T}}{2.\hat{T}}(p.T_{p}).$$

Denote $N=\frac{\hat{T}}{T_{p}}$ as the number of emitted segments for aperture synthesis, $\Delta F_{D}=\frac{1}{\hat{T}}$ as the Doppler bandwidth, $f_{Dijk}(0)=\frac{2}{\lambda_{0}}v_{rijk}(0)$ as the Doppler frequency of the scattering point, and $\hat{p}=\frac{f_{Dijk}(0)}{2.\Delta F_{D}}$ as the Doppler index, the dimensionless coordinate of the ijk-th scattering point in the projection signal plane. In the case of r=1, the first linear term can be presented as:(19)$${\hat{\Phi }}^{\left(1\right)}(0,0).(p.T_{p})=-2\pi \frac{\hat{p}.p}{N}.$$

In the case of q=1, the second linear term ${\hat{\Phi }}^{\left(1\right)}(0,0).(k.\Delta T)=\frac{d\hat{\Phi }(0,0)}{dt}(k\Delta T)$ is modified as follows. For the fast time, expression (15) is rewritten as:(20)$$\hat{\Phi }(k,p)=2\pi \frac{k}{T_{U}}\left(\frac{R_{ijk\min}(p)}{c}+k.\Delta T-\frac{R_{ijk}(p)}{c}\right).$$

Multiply and divide the right part of the expression Equation (20) with ΔT, and denote $\left(\frac{R_{ijk\min}(p)}{c}-\frac{R_{ijk}(p)}{c}\right)=-\frac{\Delta R_{ijk}(p)}{c}$, i.e.,
(21)$$\hat{\Phi }(k,p)=\frac{2\pi .k.\Delta T}{T_{U}.\Delta T}\left[-\frac{\Delta R_{ijk}(p)}{c}+k.\Delta T\right]=-\frac{2\pi .k.\Delta T}{T_{U}.\Delta T.c}\Delta R_{ijk}(p)+\frac{2\pi .{\left(k.\Delta T\right)}^{2}}{T_{U}.\Delta T}.$$

The first derivative of the phase $\hat{\Phi }(k,p)$ in respect of the fast time is written as:(22)$$\frac{d}{d\tau }\hat{\Phi }(k,p)=-\frac{2\pi .}{T_{U}.\Delta T.c}\Delta R_{ijk}(p)+\frac{4\pi .(k.\Delta T)}{T_{U}.\Delta T}.$$

For p=0,k=0, the second linear term is expressed by:(23)$${\hat{\Phi }}^{\left(1\right)}(0,0).(k.\Delta T)=\frac{d\hat{\Phi }(0,0)}{d\tau }(k\Delta T)=-\frac{2\pi .}{T_{U}.\Delta T.c}\Delta R_{ijk}(0).(k.\Delta T).$$

Denoting the number of range samples as $K=\frac{T_{U}}{\Delta T}$, the range resolution as ΔR=cΔT, and the dimensionless range coordinate of the scattering point as $\hat{k}=-\frac{\Delta R_{ijk}(0)}{\Delta R}$, the second linear term can be written as:(24)$${\hat{\Phi }}^{\left(1\right)}(0,0).(k.\Delta T)=-2\pi \frac{\hat{k}.k}{K}.$$

Based on Equations (19) and (24), the phase term Equation (15) can be written as:(25)$$\hat{\Phi }(k,p)=\Phi (k,p)-2\pi \frac{p\hat{p}}{N}-2\pi \frac{k\hat{k}}{K},$$
where Φ(k,p) is the sequence of higher-order terms.

Then, the image reconstruction Equation (14) can be expressed as:(26)$$a(\hat{k},\hat{p})=\frac{1}{N.K}\sum_{p=0}^{N-1}\sum_{k=0}^{K-1}S(k,p).\exp\left[j\left(-\Phi (k,p)+2\pi \frac{p\hat{p}}{N}+2\pi \frac{k\hat{k}}{K}\right)\right]$$
for each $\hat{p}=\bar{0,\;N-1},$$\hat{k}=\bar{0,\;K-1}$.

Based on Equation (26), it can be concluded that the image extraction is a procedure of a total phase compensation applied to the BFISAR signal. Phases defined by distances to scattering points at the moment of imaging only remain in the image. Expression (25) can be interpreted as a 2-D spatial transformation of the signal $\hat{S}(p,k)$ to the image $a(\hat{k},\hat{p})$. Operations of linear phase compensations and higher-order phase compensations can be divided. Then, Equation (26) can be rewritten as:(27)$$a(\hat{k},\hat{p})=\sum_{p=0}^{N-1}\sum_{k=0}^{K-1}S(k,p).\exp\left[-j\Phi (k,p)\right].\exp\left(j2\pi \frac{k\hat{k}}{K}\right).\exp\left(j2\pi \frac{p\hat{p}}{N}\right)$$

Based on Equation (27), stages of image extraction from the BFISAR signal can be derived.

### Basic Operations

First, based on the object’s geometry and kinematics, and the DVB-T waveform, a BFISAR signal S(k,p) reflected from the 3-D object space is created using expression (12). Second, the demodulation is executed by the multiplication of S(k,p) with the complex-conjugated waveform $\exp\left[-j.2\pi .\left(f_{0}+\Delta f\right)k.\Delta T\right]$, i.e.,
(28)$$\begin{array}{l} \hat{S}(k,p)=S(k,p).\exp\left[-j.2\pi .\left(f_{0}+k.\Delta f\right)k\Delta T\right]\\ =\sum_{i,j,k}\mathrm{rect}\left(\frac{t-t_{ijk}}{T_{U}}\right)a_{ijk}.\exp\left\{-j2\pi .\left[k.\Delta f.\tilde{t}+\left(f_{0}+k.\Delta f.\right).t_{ijk}\right]\right\}. \end{array}$$

After demodulation of the BFISAR signal, a range compression is performed by applying an inverse Fourier transform:(29)$$\tilde{S}(\hat{k},p)=K^{-1}\sum_{k=1}^{K}\hat{S}(k,p).\exp\left(j2\pi \frac{k\hat{k}}{K}\right).$$

The range-compressed BFSAR signal is azimuth-compressed by applying an inverse Fourier transform:(30)$$a(\hat{k},\hat{p})=N^{-1}\sum_{p=1}^{N}\tilde{S}(\hat{k},p).\exp\left(j2\pi \frac{p\hat{p}}{N}\right).$$

In case the obtained image is blurred, a higher-order phase correction is applied. An image contrast function or an entropy function is used to evaluate the quality of the image. The higher-order phase correction is performed by multiplication of the BFISAR signal with a phase correction function $\exp\left[-j\Phi (k,p)\right]$, i.e.,
(31)$${\hat{S}}_{c}(k,p)=\hat{S}(k,p).\exp\left[-j\Phi (k,p)\right].$$

The image reconstruction procedure is repeated with the phase-corrected signal. The coefficients of higher-order terms in the Taylor expansion of Φ(k,p) are calculated iteratively by minimizing the image cost function that evaluates the image quality. The procedure lasts until global minimum of the image cost function is achieved. The main stages of the image reconstruction algorithm are illustrated by the simulation experimental results.

## 5. Numerical Simulation Experiment

To illustrate the adequacy of the BFISAR geometrical and kinematical analysis, synthesis of the DVB-T waveform and BFISAR signal, and the algorithms for BFISAR signal formation and image reconstruction, a simulation experiment was carried out. Consider the following scenario depicted in the Cartesian coordinate system Oxyz (Figure 1). The stationary DVB-T transmitter and receiver were placed at heights near an airport or other zones under defense.

The coordinates of the DVB-T transmitter: $x^{s}=10$ m; $y^{s}=100$ m; $z^{s}=150$ m.

The coordinates of the DVB-T receiver: $x^{r}=1$ km; $y^{r}=100$ m; $z^{r}=150$ m.

Object’s trajectory parameters:

Vector velocity modulus: V=12 m/s;

Vector velocity’s guiding angles: α=π/4; β=π/4; γ=π/2.

The coordinates of the mass-center at the moment of imaging p=0:

$x_{00}(0)=100$ m; $y_{00}(0)=50$ m; $z_{00}(0)=200$ m.

A flying object, helicopter, was illuminated by DVB-T waveform segments with the following parameters:

Segment repetition period $T_{p}=1$ ms;

DVB-T symbol’s width T=0.896 ms;

DVB-T segment’s width T=1.12 ms;

DVB-T segment’s samples K=1024
*K*;

Central carrier frequency f=0.9 GHz;

DVB-T sample’s time width ΔT=0.13 μs;

Frequency bandwidth ΔF=7.61 MHz;

Number of segments for synthesis of the aperture N=256.

The object’ geometry is described in a 3-D coordinate system with a normalized intensity of scattering points in nodes $a_{ijk}=0.01$ (Figure 2). The image of the helicopter is without propellers. During the flight, propellers rotate at a high speed and can be accepted as invisible. However, it is not very correct for microwaves at a frequency of 0.9 GHz. The propellers induce additional correlation noise that was ignored in the present work.

The simulation experiment was carried out in the following order. First, the DVB-T waveform’s coefficients were calculated. Second, geometrical and kinematical parameters and distances to the object’s scattering points were calculated and sorted. Third, the BFISAR signal was created. Fourth, the image reconstruction based on the 2-D Fourier transformation realized by inverse fast Fourier transforms was implemented. The pseudo code of these stages is presented in Appendix A.

The real (a) and imaginary (b) parts of the complex BFISAR signal are presented in Figure 3. The BFISAR signal structure has properties of a complex microwave hologram.

The first step of the image reconstruction is the range compression of the BFISAR signal. The pseudo code of the range compression realized by the first inverse fast Fourier transform is presented in Appendix A. The real (a) and the imaginary parts of the complex BFISAR signal compressed in the range direction are presented in Figure 4. It can be seen that the interfering structure of the BFISAR signal occupies a particular area in the range-compressed signal.

The second step of the image reconstruction is the azimuth compression of the range-compressed BFISAR signal. The pseudo code of the azimuth compression realized by the second inverse fast Fourier transform is presented in Appendix A. The quadrature components of the final BFISAR complex image, i.e., the real (a) and imaginary (b) parts of the azimuth-compressed BFISAR signal obtained after range compression, are presented in Figure 5.

In Figure 6, the amplitude and phase of the final BFISAR complex image extracted from the DVB-T BFISAR signal by applying all steps of the imaging algorithm are depicted.

The final amplitude image of the object has a poor range and not very satisfactory azimuth resolutions. The lower range resolution is due to the narrow frequency bandwidth of the DVB-T waveform, ΔF=7.61 MHz, corresponding to a range resolution equal to 20 m. In addition, the DVB-T waveform such as a stepped frequency modulation waveform is characterized with a comparably high level of side lobes and a narrow main lobe width that can be noticed in Figure 5 and Figure 6b. The low level of scattering points’ intensities is due to the high levels of side lobes and sparse structure of the DVB-T coefficients, whose absolute values randomly accept zero values. This phenomenon is the reason the azimuth resolution to be not very satisfactory. The impact of the noise on the object’s image was not considered [43]. This important issue is the subject of future study.

## 6. Discussion

An analytical and geometrical approach was applied to describe the BFISAR scenario. Dynamic position vectors of the object’s mass-center and scattering point from the object space were derived. Based on the mathematical description of the DVB-T waveform, a model of the BFISAR signal reflected from a 3-D object was created and analyzed. BFISAR signal synthesis was interpreted as a spatial transformation of the 3-D object’s image function into a 2-D signal function with a projective phase operator. Based on the Taylor expansion of the BFISAR signal phase, linear and higher-order phase terms were defined and physically interpreted. The linear phase terms have a Fourier structure and define an object motion of first order, i.e., radial displacement. The higher-order terms define an object’s motion of higher order, i.e., tangential and radial complex maneuvering. In other words, the BFISAR signal formation is a 2-D Fourier transformation of the 3-D image function with a higher-order phase correction.

The image reconstruction is an inverse operation, i.e., an inverse spatial transformation of the 2-D signal function into a 2-D image function with an inverse projective phase operator with the same structure. Thus, the image reconstruction can be interpreted as an inverse 2-D Fourier transformation of the higher-order phase-compensated BFISAR signal.

The analytical description of the BFISAR image allows the derivation of the image reconstruction algorithm, including higher-order phase compensation of the BFISAR signal, range compression by the inverse Fourier transform in the range direction, and azimuth compression by the inverse Fourier transform in the azimuth direction of the range-compressed signal. The complex image preserves only phases defined by distances to scattering points at the moment of imaging. The higher-order terms’ coefficients are calculated iteratively using an image quality-evaluating cost function.

## 7. Conclusions

BFISAR geometrical and kinematical features, models of the waveform and signal structure, and an algorithm for extraction of the image were the focus of the present study. A BFISAR scenario with a DVB-T transmitting station of opportunity and receiver, and an object crossing baseline was analytically described by kinematical and position vector equations. The DVB-T waveform and BFISAR signal were defined by mathematical expressions. Based on the model of the BFISAR signal, an algorithm for image extraction was drawn. Transformation of the 3-D coordinates of the scattering points into their 2-D coordinates on the imaging plane was described in detail. The imaging algorithm includes the following steps: signal compression in range direction by inverse Fourier transformation, and compression in the azimuth direction by inverse Fourier transformation; the image was improved by compensation of the phases induced by object motion of higher order. To justify the correctness of the analytical and geometrical models describing the BFISAR scenario, waveform, signals, and the algorithm for extraction of the object image, graphical results of the simulation experiment were provided.

## Figures and Tables

**Figure 1 sensors-21-06662-f001:**
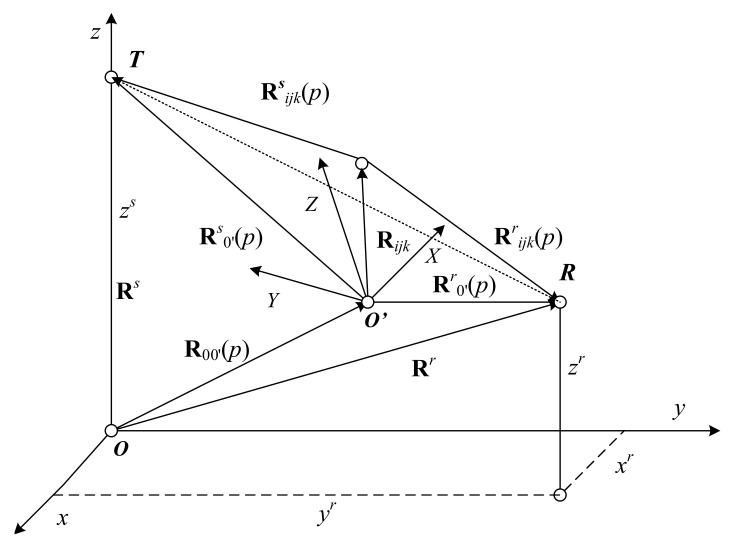
BFISAR geometry.

**Figure 2 sensors-21-06662-f002:**
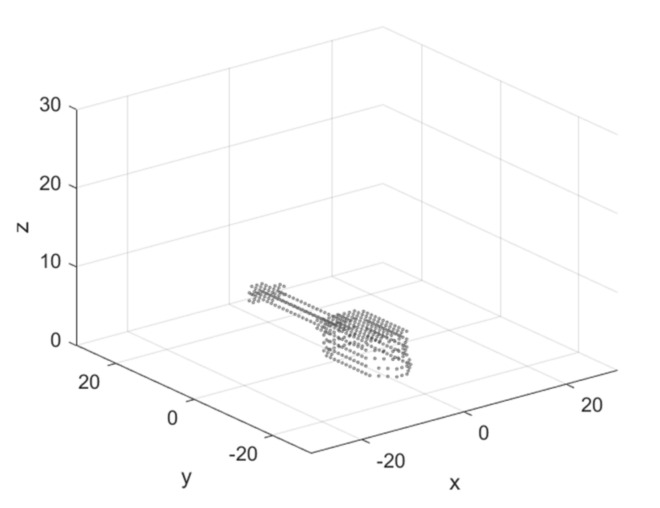
Geometry of the flying object—helicopter.

**Figure 3 sensors-21-06662-f003:**
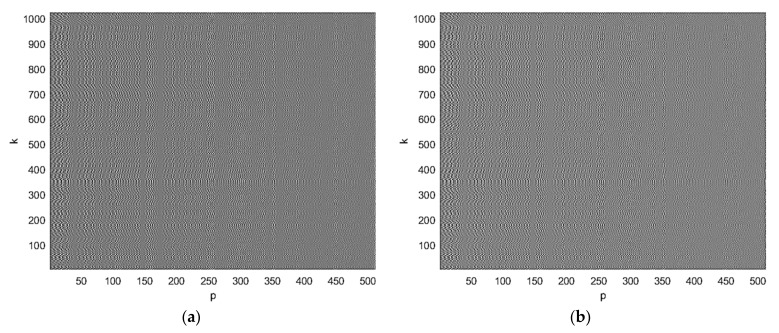
Complex BFISAR signal presented by (**a**) real and (**b**) imaginary parts.

**Figure 4 sensors-21-06662-f004:**
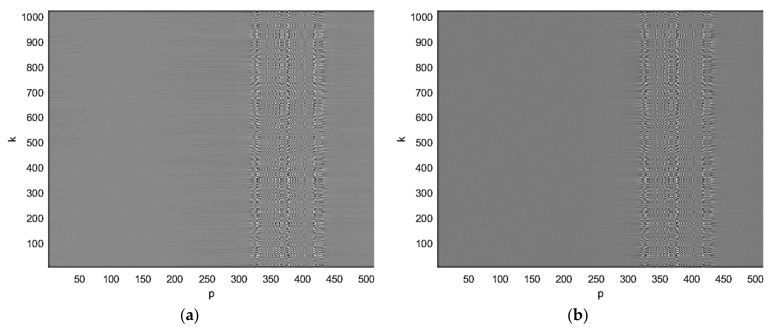
BFISAR signal after range compression: (**a**) real and (**b**) imaginary parts.

**Figure 5 sensors-21-06662-f005:**
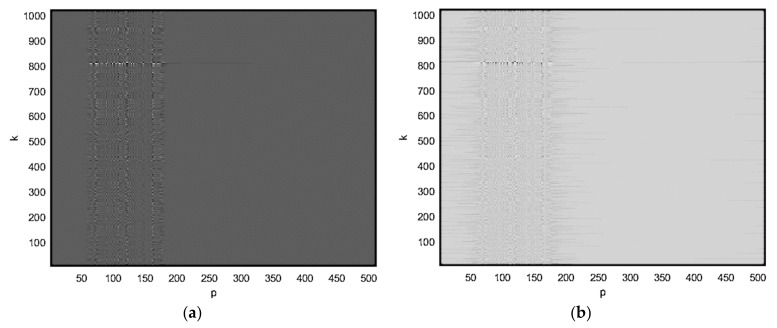
BFISAR complex image after azimuth compression and shifting: (**a**) real and (**b**) imaginary parts.

**Figure 6 sensors-21-06662-f006:**
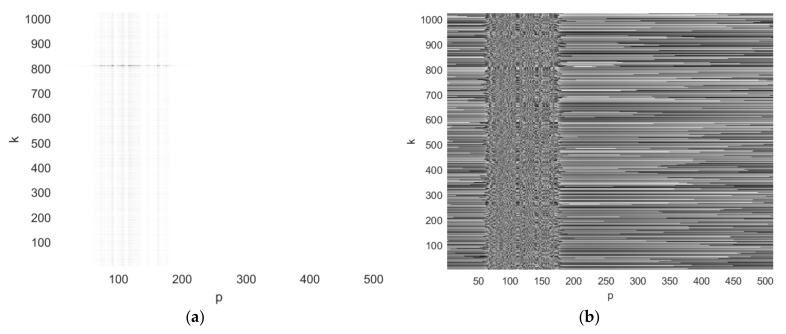
DVB-T BFISAR complex image: (**a**) amplitude image and (**b**) phase image.

## Data Availability

Not applicable.

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
