# Peer review of "Bistatic Forward ISAR with DVB-T Transmitter of Opportunity"

_sensors, 2021, doi:10.3390/s21196662_

Round 1

Reviewer 1 Report

This manuscript introduces a Bi-static Forward Inverse Synthetic Aperture Radar (BFISAR) system and its imaging model. Here are the comments:

  1. What is purpose or application field of the BFISAR?
  2. The authors underline that “spatial separated stationary transmitter, stationary receiver and an object crossing the baseline transmitter-receiver”, and the signal model is derived according to the same assumption. “Object crossing the baseline transmitter-receiver” is a strict condition, what if the target is moving over or beneath the plane of “baseline transmitter-receiver”?
  3. There is only one point-target-based experiment in section 5. Another plane-target-based experiment is required to validate the effectiveness of the proposed method.

Besides, the authors should pay more attention in the formatting and details of the manuscript, since there are too many errors throughout the manuscript. Some of the problems are listed as follows:

  1. The authors should state the full name before the abbreviation, e.g., DV-T, DVB-T, VHF, etc.
  2. The line spacing throughout the manuscript is not uniform.
  3. The language using should be improved, e.g., all the sentences in the abstract are passive and top-heavy.
  4. The symbols in figure 1 are not corresponded with the description around, e.g., symbol T and R in “transmitter T and receiver R” are not in bold, whereas T and R in figure 1 are in bold.
  5. The resolution of the figures in the experiment section are too low.
  6. Why are line 83-86 separated in different lines, and what is the meaning of the top bar.

The whole article requires carefully reviewed and polished.

Author Response

Thank you for your valueble comments and recommendation. I hope that all recommendations have been fulfilled by the authors.

Reviewer 2 Report

Please find below my detailed comments to improve the manuscript:

1) I am not satisfied with the current structure of the Introduction since it misses the important basics about radar signal processing, as well as a satisfactory review of the existing literature. Specifically, before delving into the more specific case of BFISARs and reviewing the related papers, I think it would be appropriate to provide a general introduction to the problem of radar detection. In doing so, a potential reader can be easily introduced to the topic and hopefully understand the main differences between the classical radar setup and the bistatic configuration. Since radar signal processing is experiencing a renewed interest in the last years, I suggest considering the following recent references on the topic:

- Y. Zhang et al, "Multirotors Video Synthetic Aperture Radar: System Development and Signal Processing," in IEEE Aerospace and Electronic Systems Magazine, vol. 35, no. 12, pp. 32-43, Dec. 2020

- A. Coluccia et al, "CFAR Feature Plane: A Novel Framework for the Analysis and Design of Radar Detectors," in IEEE Transactions on Signal Processing, vol. 68, pp. 3903-3916, 2020

- S. Sun et al, "MIMO Radar for Advanced Driver-Assistance Systems and Autonomous Driving: Advantages and Challenges," in IEEE Signal Processing Magazine, vol. 37, no. 4, pp. 98-117, July 2020

but authors are encouraged to add also other references they may be aware of.

2) Almost on the same line as my previous comment, it could be useful to review also some of the novel radar techniques based on machine learning and deep learning methods. This kind of method is opening the doors for a new set of unprecedented applications in the radar domain, especially exploiting signals of opportunity as done by the authors in this contribution, hence share interesting similarities. In this respect, I would suggest some quite recent references to enrich the overview provided in the Introduction:

- P. Lang et al, "A Comprehensive Survey of Machine Learning Applied to Radar Signal Processing", arXiv:2009.13702, Sep. 2020

- A. Yildirim et al, "1D Convolutional Neural Networks Versus Automatic Classifiers for Known LPI Radar Signals Under White Gaussian Noise," in IEEE Access, vol. 8, pp. 180534-180543, 2020

-G. Ricci et al, "A k-nearest neighbors approach to the design of radar detectors", Signal Processing, Vol. 174, Sept. 2020.

3) Should  R_00(p) be replaced by R_00'(p)? In the figure, I cannot see any reference to R_00(p). 

4) Line 54, R_ijk and R_00'(p) need to be inverted in the phrase.

5) The bold font is missing in eq. (2) for R_00(p).

6) A in eqs. (3) and (4) is never defined. 

7) If possible, it would be useful to add a pseudocode summarizing the main steps of the image reconstruction algorithm described in Sec. 4.

8) Can the figures be replaced with their digital version? In their current form, they do not scale very well when zooming in.

Author Response

Thank you for valuable comments and recommendations. I hape all recommendations have been fulfilled in the new version of the article.

Reviewer 3 Report

In this paper, the authors consider a BFISAR scenario, derive the motion of the target and propose an algorithm to reconstruct the target's image from the reflected signal.

I have two concerns regarding the methodology and results, and the details can be found in the references provided below to improve their manuscript.

  1. For Synthetic Aperture Radar image reconstruction, like the earlier (In)SAR and the modern B(FI)SAR alike, for real-world usage, noise plays an important role [1, 2, 3]. However, there does not seem to be any explicit modelling of noise in the author's manuscript.
  2. Although the authors' results are based on simulation, it might still be interesting for their readers to see some real-world images, like in [2].

References

[1] Mukherjee, S., Zimmer, A., Sun, X., Ghuman, P., & Cheng, I. (2020). An Unsupervised Generative Neural Approach for InSAR Phase Filtering and Coherence Estimation. IEEE Geoscience and Remote Sensing Letters, 1–5. https://doi.org/10.1109/lgrs.2020.3010504

[2] Lazarov, A. D. (2012). BISTATIC SYNTHETIC APERTURE RADAR TECHNOLOGY TOPOLOGIES AND APPLICATIONS. Proceedings of the First International Conference on Telecommunications and Remote Sensing. First International Conference on Telecommunications and Remote Sensing. https://doi.org/10.5220/0005413100030013

[3] Shi, L., Zhu, X., Shang, C., Guo, B., Ma, J., & Han, N. (2019). High-Resolution Bistatic ISAR Imaging of a Space Target with Sparse Aperture. Electronics, 8(8), 874. https://doi.org/10.3390/electronics8080874

Author Response

(The authors gave the same response as above.)

Round 2

Reviewer 1 Report

Thank you for your careful revisions. I have no further questions for the presented method. Some of the problems need to be revised before publication.

  1. Why doesn't a helicopter have a propeller in Fig.2?
  2. We can't see the image result of the target clearly in Fig.5 and Fig.6, It looks like the target is not focused?
  3. Evaluation of imaging results should be discussed, For example, the main-lobe width and the peak signal-to-noise ratio.

Author Response

Thank you for your valuable comments and recommendations. Here are answers of the posed questions.

  1. Why doesn't a helicopter have a propeller in Fig.2?
    The image of the helicopter is without propellers. During the flight, propellers rotate at a high speed and are accepted invisible. It is not very correct the propellers to be accepted invisible for microwaves at a frequency 0.9 GHz as the propeller induce additional correlation noise that is, however, ignored in the present work.

  2. We can't see the image result of the target clearly in Fig.5 and Fig.6, It looks like the target is not focused?
    The target image is focused, but the side lobe levels of the DVB-T waveform is very high around 13 dB. It does not allow the scattering points to be distinguished from the side lobes very well. An additional statement is added: In addition, the DVB-T waveform like a stepped frequency modulation waveform is characterized with a comparably high level of side lobes and narrow main lob that can be noticed in Fig. 5 and Fig. 6, b.

  3. Evaluation of imaging results should be discussed. For example, the main-lobe width and the peak signal-to-noise ratio.
    The analysis of the peak signal-to-noise ratio is a very important issue. At this stage of the problem’s study, the impact of the noise is not considered. It will be our next step of analysis and experiments and the goal of future works.
    It is added a statement:
    In addition, the DVB-T waveform like a stepped frequency modulation waveform is characterized with a comparably high level of side lobes and narrow main lobe width that can be noticed in Fig. 5 and Fig. 6, b. The low level of scattering points’ intensities is due to the high levels of side lobes and sparse structure of the DVB-T coefficients, which absolute values randomly accept zeros values.

Reviewer 2 Report

The authors correctly addressed all my comments.

Author Response

Thank you very much for your valuable comments and recommendations to improve my article.

Reviewer 3 Report

The authors have made substantial changes to improve the manuscript following the earlier round of reviews. Thus, I feel, technically, the revised version is suitable for publication in its current form. It would be advisable to get the final version of the manuscript (before publication) checked with a professional service or native speaker to rephrase some of the lengthy sentences, for example "Linear and non-linear detection algorithms with diversified robust selective properties, a highest standard of selectivity without sacrificing neither detection power under matched conditions nor CFAR property are proposed." in lines 69-71. Since this could be by subjective preference, I leave it to the best judgement of the authors.

Author Response

Answer to Reviewer 3

Reviewer:

It would be advisable to get the final version of the manuscript (before publication) checked with a professional service or native speaker to rephrase some of the lengthy sentences, for example "Linear and non-linear detection algorithms with diversified robust selective properties, a highest standard of selectivity without sacrificing neither detection power under matched conditions nor CFAR property are proposed." in lines 69-71. Since this could be by subjective preference, I leave it to the best judgement of the authors.

Thank you very much for your valuable comments and recommendations. I carefully checked my English and corrected the cited sentence.

1. Linear and non-linear detection algorithms with diversified robust selective CFAR properties are proposed.

2. It is added a statement:

In addition, the DVB-T waveform like a stepped frequency modulation waveform is characterized with a comparably high level of side lobes and narrow main lobe width that can be noticed in Fig. 5 and Fig. 6, b. The low level of scattering points’ intensities is due to the high levels of side lobes and sparse structure of the DVB-T coefficients, which absolute values randomly accept zeros values. This phenomenon is the reason of the not very satisfactory azimuth resolution. The impact of the noise on the object’s image is not considered. This important issue is the subject of future study.